# A longitudinal study of perceived stress and cortisol responses in an undergraduate student population from India

**Anuradha Batabyal[1¤], Anindita Bhattacharya[1]*, Maria Thaker[2], Shomen Mukherjee[1]**

**1** Azim Premji University, Bengaluru, India, **2** Indian Institute of Science, Bengaluru, India

¤ Current address: University of Calgary, Calgary, Canada
* anindita.bhattacharya@apu.edu.in, itsanindita@gmail.com

**Data Availability Statement:** All relevant data are within the manuscript and its Supporting Information files.

## Abstract

Young adults entering college experience immense shifts in personal and professional environments. Such a potentially stressful event may trigger multiple psychological and physiological effects. In a repeated-measures longitudinal survey (N = 6 time-points) of first year cohort of residential undergraduate students in India, this study evaluates multiple psychological parameters: PSS14 (Perceived Stress Scale), K10 (distress scale) and positive mood measures, along with salivary cortisol levels. We find that compared to women, men showed significantly lower levels of salivary cortisol and also a decrease in perceived stress (PSS14) and distress (K10) with time. By contrast, women reported similar perceived stress and distress levels over time but had higher cortisol levels at the end of the academic year. Academic stress was reported by the students to be the most important stressor. This study highlights notable gender-/sex-differences in psychological and physiological stress responses and adds a valuable longitudinal dataset from the Indian undergraduate student cohort which is lacking in literature.

## Introduction

Emerging adults at the start of university face multiple and often unique challenges during their first year. Coping with a new academic environment, relational responsibility, future financial security, and searching for their own independence and identity are all challenges that can elicit stress [1].

The major and generalised physiological reaction mechanism by which individuals cope with any of these stressors is through the activation of Hypothalamo-Pituitary-Adrenal (HPA) axis. It is well understood that secretion of glucocorticoid hormones, especially cortisol from the adrenal gland, mediates a suite of physiological responses that has immediate adaptive function to reduce the impact of the stressor [2].

Prolonged and repeated encounters with stressors, however, can lead to dysregulation of the HPA axis, resulting in detrimental effects on multiple organs and systems [3–6]. Hyper secretion of cortisol can also negatively affect the hippocampus, thereby influencing the acquisition and retainment of memories [7, 8]. Habituation of the HPA axis to repeated stressors is

**Funding:** Azim Premji University provided funding to AB and SM to conduct the study and allowed them to use their facilities. Funding for this research was supported in part by the DBT-IISc partnership program to MT. The funders had no role in study design, data collection and analysis, decision to publish, or preparation of the manuscript.

**Competing interests:** The authors have declared that no competing interests exist.

also common, and this leads to lowering of cortisol levels or blunted diurnal cortisol profiles [9]. Thus, not only does the HPA axis function in maintaining basal and stress-related homeostasis, but it also regulates emotional and cognitive centres in the brain [2, 10]. For students, chronic or repeated stress is therefore likely to interfere with their present academic performance as well as affect long term physical and mental health.

Extensive research to date has shown the interconnectivity of physiological stress reactivity and psychological perception of stress, but these studies find variable patterns [11]. In some cases, perceived control over stressful events and the perception of a stressful stimulus have been shown to affect physiological responses while in others no correlation or negative relation between physiological responses (such as cortisol levels) and psychological or subjective stress measures were found (e.g., [11]). Some of this variation and lack of a generalized pattern may be attributed to the fact that immediate responses to acute stressors are best captured by physiological measures whereas psychological variables convey overall state of mind over a longer period. Thus, to better understand and assess stressful life events for new college students, multiple psychological measures, in combination with physiological measures of cortisol would provide a more comprehensive model approach.

Stress in undergraduate students in India is considered high and puts that population of learners at high risk for mental and physical health complications or problems [12]. To understand how students, cope with the change in their academic and personal environment, in the present study we tracked a cohort of residential undergraduate students from a university in India during their entire first year of joining the academic program.

We measured psychological stress parameters and salivary cortisol in each student with a repeated sampling design, which allowed us to determine how these stress indices change for each sex during the course of the year.

We predicted differences in cortisol and perceived stress responses between men and women (similar to other findings [13, 14]). We also predicted a decrease in perceived as well as physiological stress response over time as individuals were expected to adjust to the novel academic and social environment by the end of their first year of college.

As the first longitudinal study to test both perceived and physiological stress responses in undergraduate college students from India, our study addresses an important gap in our understanding of stress reactivity during a critical transition for young adults.

## Material and methods

Written informed consent was obtained from all the participants. All the participants in the research were above 18, hence we did not have to take permission from their guardians.

### Participants

Twenty-five undergraduate residential students from the biology major participated in this longitudinal study. Participation for this study was voluntary and we had six repeated measurements for ~20 individuals across the study duration (Men = 7, Women = 15; age = 18-21years).

An initial meeting with participants after their first few weeks of joining the semester was done where the researchers explained the course of the study and a research assistant obtained informed consent. The participants completed demographic and psychological questionnaires and were then instructed about the standardized collection of saliva samples according to the study protocol. Participants were asked not to brush their teeth or eat at least 30 minutes before sampling, a note was also made about their wake time.

The study was conducted from August 2018 to May 2019 with sampling done in the months of August (1), September (2), November (3), January (4), March (5) and May (6), resulting in 3 samples in the first semester (August-November 2018) and 3 samples in the second semester (January-May 2019). The last two sampling points at the end of the second semester (March and May 2019) were during assignment submission and during the end-of-the-year examinations.

## Procedure

Physiological stress was measured as cortisol levels from salivary samples and perceived stress was assessed through questionnaires. Participants provided saliva samples between 0800–0830 h; no consumption of food or drink and brushing of teeth were completed at least 30 mins prior to providing the samples. Participants were requested to passively accumulate and provide ~1-2ml saliva in conical 10ml centrifuge tubes [15]. All samples were stored at -20˚C for further analysis (see below). On the same day of saliva collection, participants also completed questionnaires that assess perceived stress (see below). We did not control for menstrual cycle phase for the women participants as early morning cortisol responses are not expected to be significantly affected by menstrual phase [16]. We excluded one individual who was on medication as this would severely affect their cortisol response.

Sample sizes across time points: We had different sample sizes across time points and across physiological and psychological measures.

Time 1: Cortisol data: N = 23 (F = 16, M = 7); Psychological data: N = 18 (F = 11, M = 7)

Time 2: Cortisol data: N = 24 (F = 16, M = 8); Psychological data: N = 25 (F = 16, M = 9)

Time 3: Cortisol data: N = 21 (F = 16, M = 5); Psychological data: N = 23 (F = 15, M = 8)

Time 4: Cortisol data: N = 25 (F = 16, M = 9); Psychological data: N = 24 (F = 17, M = 7)

Time 5: Cortisol data: N = 24 (F = 17, M = 7); Psychological data: N = 24 (F = 17, M = 7)

Time 6: Cortisol data: N = 23 (F = 16, M = 7); Psychological data: N = 23 (F = 16, M = 7)

## Salivary cortisol

Before cortisol analysis, all samples were thawed and centrifuged at 3500rpm for 20 min and the supernatant was used for further analyses. Enzyme-Immuno Assay kits (Arbor Assay DetectX Cortisol K003-H5) were used to measure circulating cortisol level. EIA kits were first optimized (as per [17]) and we subsequently analysed samples at a dilution ratio of 1:4 in duplicate across 4 assays. Percent recovery of cortisol in the assay was 98.93, with an intra-assay coefficient of variation of 0.12–6.84 and an inter-assay coefficient of variation at 9.51 (Inter-assay CV were calculated from a lab standard of known concentration placed on all plates). Hormone levels were determined in reference to seven-point standard curve with a limit of detection at 0.016 ng/ml for cortisol.

## Psychological evaluation

All participants completed questionnaires on the same day as saliva collection. Three self-reported subjective measures of psychological state were quantified from the questionnaire data: K10 distress scale [18], Perceived Stress Scale-14 [19] and positive mood measure [20]. K10 (or distress) was calculated from answers to 10 questions that describe how often the individual feels tired / nervous / distressed, with scores that range on a 5-point scale from "all of

the time" to "none of the time". Positive mood scores were calculated similarly on a 5- point scale (ranging from "extremely" to "none at all") derived from 13 questions that assess how inspired / peaceful / satisfied individual's felt over the last month. The perceived stress scale (PSS) was derived from a 14-item questionnaire with scores ranging from 0–4 that capture the frequency of both positive and negative feelings over a monthly duration. The positive scores were reversed before calculating the final PSS score [19].

Along with the above three measures, we also asked participants one open ended question: "What aspects of their life caused maximum stress in the last month: academic, own health, health of close one, relationship stress, family issues, financial issues or any other".

During each of the sampling time-points, general data on health issues and medical history were also obtained. Samples of the study questionnaire is provided in the (S1 File).

## Ethical consideration

Participation in this study was voluntary, and an informed consent form was signed by participants at all time points during the sampling. The study was performed in accordance with the Declaration of Helsinki and was approved by the ethics committee of the University. Institutional Review Board, Azim Premji University, Bangalore, India approved the study. IRB Approval Reference: IRB/SLS/Small Grant/December 17, 2018/Stress & Cortisol Level.

## Statistical analyses

Individuals were sampled at 6 time points and cortisol, PSS14, K10 and Mood scores were quantified at each. To test whether physiological (Cortisol) and psychological (K10, Mood, PSS14) parameters were different between sexes across the six different time points, we performed separate linear mixed effect model analyses (R package: lmer and lmer Test, [21]) for all response variables except for the K10 Distress scale where we used a generalised linear mixed effect model as data was non-normal (R package: glmmADMB, [22]). We also log-transformed the cortisol data to normalise it before statistical analysis. In all models, sex and time points were used as interacting fixed factors and individual identity as a random effect.

We scored presence and absence of various types of stressors (based on the open-ended question): academic, own health, health of close one, relationship stress, family issues and financial issues and used a generalised linear mixed effect modelling with a binomial distribution to test which stressor type contributed most across each time point for both sexes. For this, data was divided across sexes and we ran two separate GLMMs, with presence or absence (1/0) as our response, time point and stressor type as our fixed factors and individual identity as random effect. All post-hoc comparisons were performed using lsmeans function (package: lsmeans; [23] and all statistical analyses was performed using R version 3.6 (R core team 2019).

## Results

### Psychological and physiological stress across time

In the current sample we did not have any students who reported any depressive symptoms or suicidal ideation over the study period. Patterns of K10 Distress scores showed a significant interaction effect of sex and time point, wherein women were not different across all time points (all $t<2.50$, $p>0.05$, Fig 1A, Table 1) but men showed decreased levels of distress from time points 1 to 4 ($t = 3.86$, $p = 0.002$, Fig 1A, Table 1), 1 to 6 ($t = 2.94$, $p = 0.044$, Fig 1A, Table 1) and also from 2 to 4 ($t = 2.98$, $p = 0.039$, Fig 1A, Table 1).

Similarly, for PSS14, we found no difference across time points for women (all $t<1.85$, $p>0.05$, Fig 1B, Table 1) but men showed a decrease in PSS14 scores from time points 1 to 5

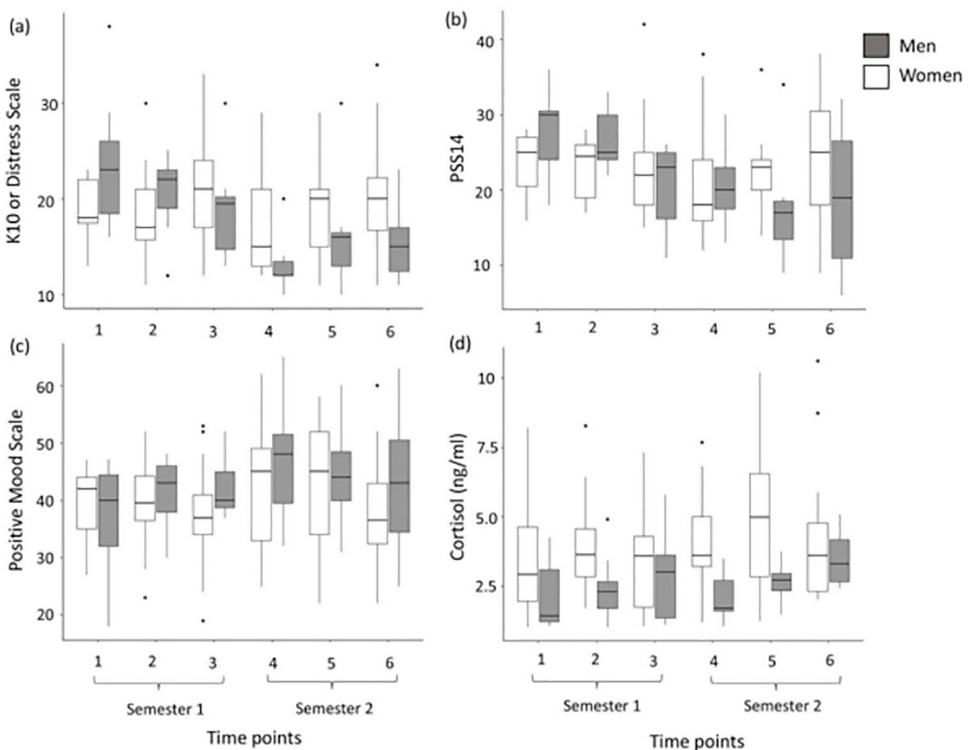

**Fig 1. Psychological measures and physiological measures.** (a) K10 Distress scale, (b) PSS14, (c) Positive mood scale and physiological measures of (d) Cortisol levels across six time points in the academic year. Grey and white boxes represent responses of men and women respectively. Boxplots show medians, quartiles, 5th and 95th percentiles and extreme values.

($t = 2.96$, $p = 0.046$, Fig 1B, Table 1) and 2 to 5 ($t = 2.98$, $p = 0.041$, Fig 1B, Table 1). There was no significant difference between the sexes or across time points for Mood scores (all t<1.41, p>0.1, Fig 1C, Table 1). The contribution of individual identity (random effect) across all models was 1.5 standard deviation or lower.

**Table 1. Mean ± SE of all psychological (K10, Mood, PSS14) and physiological (cortisol) parameters for both sexes.**

| Sex | Time point | K10 | Mood | PSS14 | Cortisol (ng/ml) |
|---|---|---|---|---|---|
| Women | 1 | 19.18±0.98 | 39.00±1.92 | 23.63±1.26 | 3.47±0.46 |
| | 2 | 18.12±1.16 | 39.00±1.88 | 22.93±0.98 | 3.96±0.41 |
| | 3 | 21.40±1.66 | 37.46±2.38 | 23.26±1.84 | 3.22±0.44 |
| | 4 | 17.17±1.22 | 43.05±2.70 | 20.82±1.81 | 4.07±0.44 |
| | 5 | 18.55±1.08 | 42.29±2.58 | 22.58±1.50 | 4.98±0.66 |
| | 6 | 20.62±1.55 | 37.81±2.42 | 24.87±2.09 | 4.17±0.60 |
| Men | 1 | 23.71±2.92 | 36.85±4.04 | 27.57±2.33 | 2.19±0.51 |
| | 2 | 20.44±1.33 | 41.11±2.12 | 26.55±1.37 | 2.45±0.43 |
| | 3 | 19.00±1.90 | 42.00±1.77 | 20.25±2.19 | 2.97±0.84 |
| | 4 | 13.28±1.20 | 46.71±4.29 | 20.57±2.09 | 2.11±0.28 |
| | 5 | 16.42±2.44 | 44.57±3.50 | 17.71±3.09 | 2.64±0.26 |
| | 6 | 15.42±1.55 | 43.00±4.86 | 18.85±3.77 | 3.49±0.40 |

Cortisol levels also differed between the sexes across time. For cortisol, men had significantly lower levels of circulating cortisol compared to women across all time points ($t$ = -2.05, $p$ = 0.042, Fig 1D, Table 1). For women only, cortisol levels were also significantly higher at time point 5 compared to their first time point ($t$ = 2.22, $p$ = 0.028, Fig 1D, Table 1).

## Types of stressors

All students reported binary (yes/no) responses for the presence or absence of different types of stressors: academic, own health, health of close ones, relationship, financial and family. There were no gender-/sex-differences in the types of stressors reported (z = -0.66, p = 0.509), but different type of stressors significantly differed across time. We thus divided the data across sexes and performed separate mixed effect models to understand how the stressors were different across time points.

Total number of stressors reported by both men and women were lowest at time point 4 compared to time point 1 (men: z = -2.5, p = 0.012, Fig 2A; women: z = -2.86, p = 0.004, Fig 2B). Additionally, men also reported a significantly lower number of stressors at time point 5 compared to 1 (z = -2.23, p = 0.025, Fig 2A).

Academic stress was reported by both males and females more often than any other stressor type (all z>3, p<0.03, Fig 2). "Own health" was the second most frequently reported source of stressor, compared to financial stress which was reported the least number of times (men: z = 3.02, p = 0.026, Fig 2A; women: z = 3.04, p = 0.024, Fig 2B). Men also reported relationship stress more often than financial stress (z = 2.90, p = 0.037, Fig 2A).

## Discussion

We provide a rare longitudinal view of stress in residential undergraduate students from India, using a combination of psychological and physiological assessments. From our year-long assessment, we obtained detailed data of how students perceive stress as well as how their baseline cortisol levels change during their first year of university.

Two of the sampling points in our study were during assignment submission periods and we find an increase in salivary cortisol during one of these time points (time 5, Fig 1D) compared to the first sample when there was no immediate academic pressure. This elevation in cortisol is likely to be attributed to the anticipation of the assignment deadlines as stressful events, which is similar to other studies where students are found to increase to comparable cortisol levels during examination [24, 25].

One of the most interesting results from our study is the gender-/sex-difference, where we find that women in an undergraduate program in India have higher salivary cortisol levels compared to men during their first year of college which was different from other stress studies that have either found no significant difference between the sexes or have found that young men have elevated cortisol levels compared to young women when challenged with acute stressful tasks, such as examinations or other laboratory stress tests (see review in [26]).

Men in our study cohort also showed a decrease in both perceived stress score (PSS14) and distress score (K10) with time. During the second semester, men reported significantly lower scores for perceived stress whereas women did not change their perception of stress from the start of the semester to the end of the academic year. This suggests that men might be adjusting to the new academic environment faster than women. These results are similar to several studies where women report perceiving higher stress than men in college [27, 28]. It has also been observed in research studies that women belonging to South East Asian countries report more mental distress and socio-cultural factors contribute immensely to these [29].

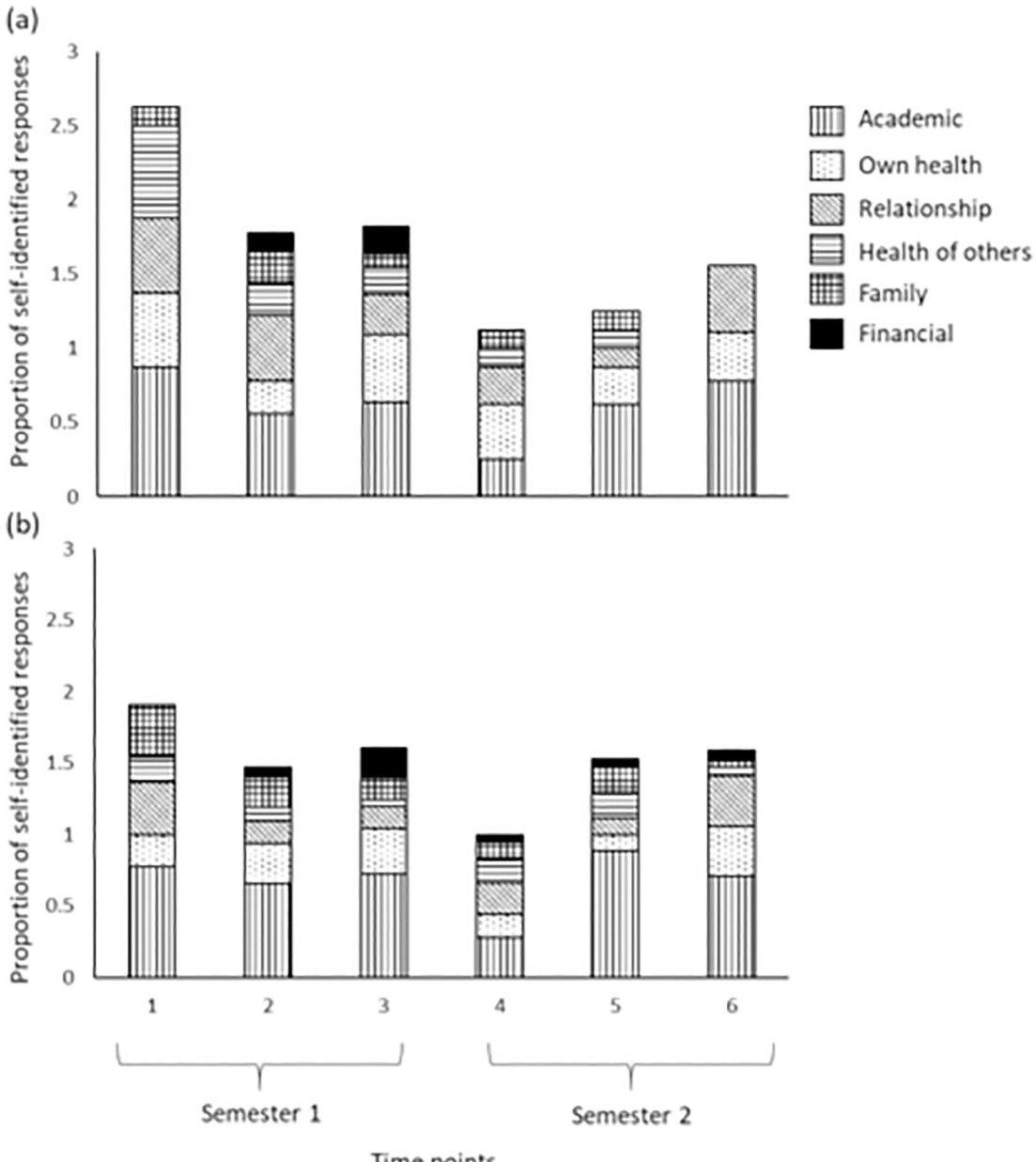

**Fig 2. Stacked bar plot for proportion of self-identified responses for types of stressors across time.** (a) In men and (b) women undergraduate student participants.

Additionally, coping strategies also differ between men and women. Men typically rely on problem-solving strategies, whereas women tend to use emotion-focussed coping strategies, which require social and emotional support and acceptance [30]. In a new academic environment, perceived stress and the type of coping strategy that depends on other external factors might be the reason for women showing higher overall and prolonged stress levels than men. Irrespective of the perception of stress by women and men, cortisol levels were highest for women during assignment submission and the end-of-the-year- examination time compared to the start of the semester. This further strengthens the view that cortisol levels are majorly

influenced by immediate stressors compared to perceived stress measures which are representation of a much longer time period.

The major contributing factor for perceived stress in our study was academic pressure as that was reported most often by both men and women across all time points (similar to [31, 32]). Overall stressors reported was lowest at the beginning of semester 2, when academic pressure was low, and when students returned to classes after a break. Though a similar pattern was expected at the beginning of semester 1, the anticipation of a new academic environment and the shift from high school to a university environment likely led to a heightened perception of overall stress at the beginning of the first semester (similar to [33, 34]). Other studies have found that financial stress is a major concern for undergraduates, especially those coming from different socio-economic backgrounds [35, 36]. In our study, financial stress is of least concern, which is most likely due to the financial security provided by the University through need based scholarships (that waves tuition fee & hostel rent, and also covers student food expenses), and the fact that students stay within a residential campus and are therefore partially protected from the daily exposure of individual and familial financial stress (see [37, 38]).

Stress has become a pertinent issue among youth globally but particularly so among Asians. Reviewing the literature on self-harm among Asians, Thompson and Bhugra [39] proposed a theoretical model which suggests that high parental pressure affects perceptions of stress that could lead to deliberate self-harm among Asian adolescents. They described Indian cultural expectations to succeed and claimed that the there's an excessive focus on academic and economic success. Moreover, there's a clear stigma attached to failure, acceptance of authority of elders and an unquestioning compliance from younger family members which could lead to internal conflicts.

In India, the age specific suicide rate among 15–29 year is on the rise increasing from 3.73 to 3.96 per 100,000 populations per year from 2002 to 2011 according to National Crime Record Bureau. According to an article by Dandona and colleagues [40], suicide was the leading cause of death in India in 2016 for those aged 15–39 years; 71.2% of the suicide deaths among women and 57.7% among men were in this age group. Based on these alarming figures, stress among Indian youth is clearly an important concern.

As the first study of student stress at this undergraduate program, we intentionally minimized the intrusion on student's lives. This meant that we had some limitations. For example, we did not obtain information on smoking behaviour or the consumption of drugs or alcohol. We also did not include any questions on the menstrual cycle phase for women, because previous studies report that morning cortisol responses are not altered by menstrual cycle phase [16, 41]. We also intentionally excluded information on diet, as our study was on a residential campus, and thus all college students were provided the same food during the entire study duration. We also had no control over the sleep cycle and whether students experienced a sound sleep or the exact duration of sleep either. Wake time of most students were within an hour before the sample collection (we asked wake time for some students during sample collection). We understand the data limitation here as we would not know where in the cortisol daily cycle we are actually sampling. However, we think that the repeated sampling design might have taken care of this limitation to a large extent.

Given the voluntary nature of our study and the stochasticity of sex ratios, we had more women than men in our study group, we also had a relatively small sample size because of the cost and time associated with taking a larger sample, cost of salivary cortisol kits, and lab analysis in a limited institutional budget in an applied setting. Future studies may benefit from multiple daily samplings for measuring salivary cortisol to further enhance the robustness of the measure. Another significant limitation of our study is that our findings may not be generalized to all college students because of the small convenience sample and unmeasured

confounders (e.g., personality types, lifestyle factors and social factors) which may impact the results, especially when examining the gender differences.

Despite these potential weaknesses, the importance of measuring both physiological and psychological measures are clear, as we observe different patterns of cortisol and perceived stress across different time points in the academic semester. Future research using multiple variables for stress measurements with a greater sample size and longitudinal design would help in addressing the growing issue of poor mental health in academia, in India and globally. With such information on stress-induced triggers and when to expect them, we can also design optimal intervention strategies for a healthy young adult population.

## Supporting information

**S1 File. Questionnaires used in the study.**
(PDF)

**S1 Data.**
(XLSX)

## Acknowledgments

We would like to thank the 2018 BSc Biology students of Azim Premji University for volunteering for this study. We would also like to thank Anagha Menon and Elizabeth Matthew for help with collecting saliva samples.

## Author Contributions

**Conceptualization:** Anindita Bhattacharya, Shomen Mukherjee.

**Data curation:** Anuradha Batabyal.

**Formal analysis:** Anuradha Batabyal, Maria Thaker.

**Funding acquisition:** Anindita Bhattacharya, Shomen Mukherjee.

**Investigation:** Anuradha Batabyal.

**Methodology:** Anindita Bhattacharya, Shomen Mukherjee.

**Project administration:** Anindita Bhattacharya, Shomen Mukherjee.

**Supervision:** Anindita Bhattacharya, Shomen Mukherjee.

**Writing – original draft:** Anuradha Batabyal.

**Writing – review & editing:** Anindita Bhattacharya, Maria Thaker, Shomen Mukherjee.

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
