## [Decision Letter · Decision Letter 0]

22 Jan 2021

PONE-D-20-29949

A longitudinal study of perceived stress and cortisol responses in an undergraduate student population from India

PLOS ONE

Dear Dr. Bhattacharya,

Thank you for submitting your manuscript to PLOS ONE. After careful consideration, we feel that it has merit but does not fully meet PLOS ONE’s publication criteria as it currently stands. Therefore, we invite you to submit a revised version of the manuscript that addresses the points raised during the review process.

We look forward to receiving your revised manuscript.

Kind regards,

Julia Dratva

Academic Editor

PLOS ONE

Additional Editor Comments:

I believe the research was well done and is a first "proof of feasibility"of longitudinal cortisol measures generating first hypotheses in the Indian undergraduate context. The longitudinal study design partly conteracts the small sample size, however, it remains a concern.

In addition, to the reviewers' comments please address the following:

Throughout the article the term "sex difference" is used. I suggest to use gender-difference, since the authors themselves point to non-biological factors potentially causing the increase in cortisol in women. Or gender-/sex-difference if they which to point to potential biological differences in the stress-response by sex.

Impact of "stressors" were assessed in stratified analyses over all time points. As far as I understand they did not adress interactions of sex and stressors.

Please adress the limitations in more detail: sample size, missing confounders; as well as the validity of cortisol measures. The authors should also dicuss the generalizability of the results and how unmeasured confounders might impact the results, especially from a gender point of view.

Reviewers' comments:

Reviewer's Responses to Questions

**Comments to the Author**

1. Is the manuscript technically sound, and do the data support the conclusions?

Reviewer #1: Yes

Reviewer #2: No

2. Has the statistical analysis been performed appropriately and rigorously? 

Reviewer #1: Yes

Reviewer #2: Yes

3. Have the authors made all data underlying the findings in their manuscript fully available?

Reviewer #1: Yes

Reviewer #2: Yes

4. Is the manuscript presented in an intelligible fashion and written in standard English?

Reviewer #1: Yes

Reviewer #2: Yes

5. Review Comments to the Author

Reviewer #1: The authors present a very interesting and well written longitudinal study on perceived psychological stress and cortisol levels in a student population. I very much enjoyed to review this manuscript. I have only some minor points which should be addressed by the authors.

A. Participants (page 5, lines 91-101):

A1. The authors should provide some further information on the selection process, i.e. sampling procedure. How were students recruited?

A2. The authors state that 25 students participated in the study and six measurements were available for ~20 individuals. Please indicate whether patients were included and analyzed on an intention-to-treat basis and if so how many observations were present at each time point. If only individuals with complete measurements were included, please justify. What do the authors mean by ~ (approximately) 20 individuals? Please specify precisely for how many subjects measurements were available at all time points.

A3. Please justify your sample size.

B. Discussion:

B1. (page 13, line 285): There is a typo in the suicide incidence rate, i.e. “1,00,000” should probably read “100,000”.

Reviewer #2: The study tracked psychological (distress, perceived stress, positive mood) and salivary cortisol responses over an academic year in 25 residential undergraduate students in India. The study reported sex differences in stress (K10; PSS; men decreased over time while women remained stable) and cortisol responses (overall higher for women, and specifically higher at the end of the academic year).

Strengths of the study included the longitudinal assessment over the 1-year study period and the recruitment of an Indian undergraduate student cohort.

Major limitations are listed below.

First and foremost, a major limitation is the small sample size: n = 25 total with nMen = 7, nWomen = 15 across the study period. The uneven and small sample size for men (n= 7) is particularly of concern given that the main results focus on sex differences.

My second concern is the measurement of cortisol, which was obtained at 0800-0830 h. Without more detailed information on wake time and sleep, it is unclear what a cortisol value at this specific time means. Differences in wake time need to be taken into account with this data.

Other Comments:

The authors review evidence for the increased risk for suicide in Asian young adults, but did not include information regarding depressive symptoms or suicidal ideation over the study period. If available, such data could strengthen the paper.

How did the authors ensure that no consumption of food/drink or teeth brushing was conducted half an hour before providing the salivary sample?

The authors name “self-imposed” stress as a reason for higher overall and maintained stress levels, but also mention high parental pressure. I would suggest rephrasing this word, since self-imposed implies that there is no external/social pressure. I might also add that social/cultural expectations to succeed might be particularly prominent for young women.

6. PLOS authors have the option to publish the peer review history of their article (what does this mean?). If published, this will include your full peer review and any attached files.

Reviewer #1: **Yes: **Thomas Volken

Reviewer #2: No

---

## [Author Response · Author response to Decision Letter 0]

14 Apr 2021

Dear Editor and Reviewers, 

Please find responses to your comments below:

Responses to Editor’s comments:

Throughout the article the term "sex difference" is used. I suggest to use gender-difference, since the authors themselves point to non-biological factors potentially causing the increase in cortisol in women. Or gender-/sex-difference if they which to point to potential biological differences in the stress-response by sex. 

Point well taken, in the revised manuscript gender/sex difference has been used. 

Impact of "stressors" were assessed in stratified analyses over all time points. As far as I understand they did not address interactions of sex and stressors.

We did look at sex and stressors interaction, but it was not significant. Please see lines 217-226:

There were no sex differences in the types of stressors reported (z=-0.66, p=0.509), but different type of stressors significantly differed across time. We thus divided the data across sexes and performed separate mixed effect models to understand how the stressors were different across time points. 

Total number of stressors reported by both men and women were lowest at time point 4 compared to time point 1 (men: z=-2.5, p=0.012, Fig. 2a; women: z=-2.86, p=0.004, Fig. 2b). Additionally, men also reported a significantly lower number of stressors at time point 5 compared to 1 (z=-2.23, p=0.025, Fig. 2a).

Please address the limitations in more detail: sample size, missing confounders; as well as the validity of cortisol measures. The authors should also discuss the generalizability of the results and how unmeasured confounders might impact the results, especially from a gender point of view.

These points have been addressed in the concluding paragraph (lines 314-326) of the revised manuscript. 

Responses to the Reviewers’ comments:

Reviewer 1:

A. Participants (page 5, lines 91-101):

A1. The authors should provide some further information on the selection process, i.e. sampling procedure. How were students recruited?s

An initial meeting with participants after their first few weeks of joining the semester was done where the researcher explained the course of the study and a research assistant obtained informed consent. The participants completed demographic and psychological questionnaires and were then instructed about the standardized collection of saliva samples according to the study protocol. Participants were asked not to brush their teeth or eat at least 30 minutes before sampling (lines 98-103)

A2. The authors state that 25 students participated in the study and six measurements were available for ~20 individuals. Please indicate whether patients were included and analyzed on an intention-to-treat basis and if so how many observations were present at each time point. If only individuals with complete measurements were included, please justify. What do the authors mean by ~ (approximately) 20 individuals? Please specify precisely for how many subjects measurements were available at all time points.

Sample sizes across time points: We had different sample sizes across time points and across physiological and psychological measures. 

Time 1: Cortisol data: N=23 (F=16, M=7); Psychological data: N=18 (F=11, M=7)

Time 2: Cortisol data: N=24 (F=16, M=8); Psychological data: N=25 (F=16, M=9)

Time 3: Cortisol data: N=21 (F=16, M=5); Psychological data: N=23 (F=15, M=8)

Time 4: Cortisol data: N=25 (F=16, M=9); Psychological data: N=24 (F=17, M=7)

Time 5: Cortisol data: N=24 (F=17, M=7); Psychological data: N=24 (F=17, M=7)

Time 6: Cortisol data: N=23 (F=16, M=7); Psychological data: N=23 (F=16, M=7)

This information has been added in the text (lines – 124-129 )

A3. Please justify your sample size. 

To the best of our knowledge, this is the first study of its kind in the Indian population and was a preliminary investigation looking at the correlation between physiological and psychological correlates of stress. We started with 34 individuals. However, since the batch had a low number of male students, we could not increase our male sample size. The initial number of males was 9, and even at a later time, 8-9 males continued to participate in the study, indicating that most females dropped out. Though the sample size is small, the total salivary samples were 140. It was also quite challenging to follow up with students for 6 points of time (1 year). The cost and time associated with taking a larger sample, buying salivary cortisol kits, and analysing them in the lab were other factors that determined our sample size. These are some limitations of our current study (which we have now added in detail), but we hope to address these in future studies. 

B. Discussion:

B1. (page 13, line 285): There is a typo in the suicide incidence rate, i.e. “1,00,000” should probably read “100,000”.

Thanks for pointing this out, we have made the change in the revised manuscript

Reviewer 2:

First and foremost, a major limitation is the small sample size: n = 25 total with nMen = 7, nWomen = 15 across the study period. The uneven and small sample size for men (n= 7) is particularly of concern given that the main results focus on sex differences.

This concern has been addressed in the response given to the first reviewer justifying the small sample size. Please see lines (51-61) above.

My second concern is the measurement of cortisol, which was obtained at 0800-0830 h. Without more detailed information on wake time and sleep, it is unclear what a cortisol value at this specific time means. Differences in wake time need to be taken into account with this data.

We had no control over the sleep cycle and whether students experienced a sound sleep or the exact duration of sleep either. Wake time of most students were within an hour before the sample collection (we asked wake time for some students during sample collection). We understand the data limitation here as we would not know where in the cortisol daily cycle we are actually sampling. However, we think that the repeated sampling design might take care of this limitation to a large extent. 

The authors review evidence for the increased risk for suicide in Asian young adults but did not include information regarding depressive symptoms or suicidal ideation over the study period. If available, such data could strengthen the paper.

In the current sample we did not have any students who showed depressive symptoms or suicidal ideation over the study period. (lines- 188-189) in the revised manuscript.

How did the authors ensure that no consumption of food/drink or teeth brushing was conducted half an hour before providing the salivary sample? 

We instructed students specifically on this criterion and confirmed it before sample collection. This has been added in the methods ( lines 98-103) in the revised manuscript.

The authors name “self-imposed” stress as a reason for higher overall and maintained stress levels, but also mention high parental pressure. I would suggest rephrasing this word, since self-imposed implies that there is no external/social pressure. I might also add that social/cultural expectations to succeed might be particularly prominent for young women.

This is a relevant point; the word has been rephrased and a relevant research study has been added in the revised manuscript. (please see lines 263-266)

---

## [Decision Letter · Decision Letter 1]

11 May 2021

PONE-D-20-29949R1

A longitudinal study of perceived stress and cortisol responses in an undergraduate student population from India

PLOS ONE

Dear Dr. Bhattacharya,

Thank you for submitting your manuscript to PLOS ONE. After careful consideration, we feel that it has merit but does not fully meet PLOS ONE’s publication criteria as it currently stands. Therefore, we invite you to submit a revised version of the manuscript that addresses the points raised during the review process.

We look forward to receiving your revised manuscript.

Kind regards,

Julia Dratva

Academic Editor

PLOS ONE

Journal Requirements:

Additional Editor Comments (if provided):

Thank you for addressing the reviewer's points and requests for improvement.

One last point, should still be considered, you answered the reviewer 2 point on cortisol measurement (see below), however, you did not take it up in the discussion. It seems a limitation worth mentioning, especially as you did ask for wake time (add to methods):

"We had no control over the sleep cycle and whether students experienced a sound sleep or the exact duration of sleep either. Wake time of most students were within an hour before the sample collection (we asked wake time for some students during sample collection). We understand the data limitation here as we would not know where in the cortisol daily cycle we are actually sampling. However, we think that the repeated sampling design might take care of this limitation to a large extent. "

Please introduce this limitation and your arguments into the manuscript.

Reviewers' comments:

Reviewer's Responses to Questions

**Comments to the Author**

1. If the authors have adequately addressed your comments raised in a previous round of review and you feel that this manuscript is now acceptable for publication, you may indicate that here to bypass the “Comments to the Author” section, enter your conflict of interest statement in the “Confidential to Editor” section, and submit your "Accept" recommendation.

Reviewer #1: All comments have been addressed

2. Is the manuscript technically sound, and do the data support the conclusions?

Reviewer #1: Yes

3. Has the statistical analysis been performed appropriately and rigorously? 

Reviewer #1: Yes

4. Have the authors made all data underlying the findings in their manuscript fully available?

Reviewer #1: Yes

5. Is the manuscript presented in an intelligible fashion and written in standard English?

Reviewer #1: Yes

6. Review Comments to the Author

Reviewer #1: Thank you for addressing my comments. The manuscript has been substantially improved. Especially study limitations are now much clearer.

7. PLOS authors have the option to publish the peer review history of their article (what does this mean?). If published, this will include your full peer review and any attached files.

Reviewer #1: **Yes: **Thomas Volken

---

## [Author Response · Author response to Decision Letter 1]

13 May 2021

Dear Editor and Reviewers, 

Please find responses to your comments below:

Responses to Editor’s comments:

Editor: Thank you for addressing the reviewer's points and requests for improvement.

One last point, should still be considered, you answered the reviewer 2 point on cortisol measurement (see below), however, you did not take it up in the discussion. It seems a limitation worth mentioning, especially as you did ask for wake time (add to methods):

"We had no control over the sleep cycle and whether students experienced a sound sleep or the exact duration of sleep either. Wake time of most students were within an hour before the sample collection (we asked wake time for some students during sample collection). We understand the data limitation here as we would not know where in the cortisol daily cycle we are actually sampling. However, we think that the repeated sampling design might take care of this limitation to a large extent. "

Please introduce this limitation and your arguments into the manuscript.

Response: a line [ lines 103- 104] on wake time has been added to the method section in the main manuscript. 

Data limitation about cortisol measurement not knowing the wake time has been included in the discussion section as suggested [lines 318-324] in the manuscript.

---

## [Editor Report · Decision Letter 2]

19 May 2021

A longitudinal study of perceived stress and cortisol responses in an undergraduate student population from India

PONE-D-20-29949R2

Dear Dr. Bhattacharya,

We’re pleased to inform you that your manuscript has been judged scientifically suitable for publication and will be formally accepted for publication once it meets all outstanding technical requirements.

Kind regards,

Julia Dratva

Academic Editor

PLOS ONE
---

## [Editor Report · Acceptance letter]

26 May 2021

PONE-D-20-29949R2 

A longitudinal study of perceived stress and cortisol responses in an undergraduate student population from India 

Dear Dr. Bhattacharya:

I'm pleased to inform you that your manuscript has been deemed suitable for publication in PLOS ONE. Congratulations! Your manuscript is now with our production department. 

Kind regards, 

on behalf of

Dr. Julia Dratva 

Academic Editor

PLOS ONE